# REGULARITY EXPLAINS EMERGENCE

## ABSTRACT

We investigate the mechanism behind emergence in large language models from the viewpoint of the regularity of the optimal response function $f^*$ on the space of prompt tokens. Based on theoretical justification, we provide an interpretation that the derivatives of $f^*$ are in general unbounded and the model gives up reasoning in regions where the derivatives are large. In such regions, instead of predicting $f^*$, the model predicts a smoothified version obtained via an averaging operator. The threshold on the norm of derivatives for regions that are given up increases together with the number of parameters $N$, causing emergence. The relation between regularity and emergence is supported by experiments on arithmetic tasks such as multiplication and summation and other tasks. Our interpretation also sheds light on why fine-tuning and Chain-of-Thought can significantly improves LLM performance.

## 1 INTRODUCTION

In large language models (LLMs), emergent abilities are skills or behaviors that manifest unexpectedly when models are scaled up in size or trained on more data. These abilities often appear without being explicitly programmed, seemingly "emerging" as a result of the model's scale and complexity. They were first observed in the GPT-3 family of models (Kaplan et al., 2020), (Brown et al., 2020), sparking significant interest in research. Numerous studies have since explored these abilities across various tasks.

Emergence ability's key feature is "unpredictability" (Wei et al., 2022a). If we plot performance as a function of parameters, then at some point of scaling, performance improvement is significant, and is unpredictable from its small scale behavior. Emergence is also known to be task dependent. Ganguli et al. (2022) noted that "performance on a specific task can sometimes emerge quite unpredictably and abruptly at scale".

Three primary factors that affect emergence ability are: computation ability, number of parameters, and training dataset size (Kaplan et al., 2020; Hoffmann et al., 2024). The interaction between these factors seems to be complex. For instance, "emergence may occur with less training compute or fewer model parameters for models trained on higher-quality data" (Wei et al., 2022a). In this paper, we will focus on the role of number of parameters.

The emergence of advanced abilities in AI models has sparked crucial discussions around AI safety and controllability. Ensuring the predictability of AI systems is vital for their responsible deployment. Understanding the conditions under which emergence happens and the mechanisms behind it is a central focus of research. It offers insights into both model capabilities and limitations. Recently in (Schaeffer et al., 2023), researchers argued that emergent abilities may stem from the choice of evaluation metrics rather than fundamental changes in model behavior as scale increases.

We propose an alternative mechanism to explain the emergence of abilities in large language models. For task with unique answers from the same pool of tokens, such as arithmetic operation on decimal numbers, the complexity measured in terms of cross entropy are the same, however certain tasks are less likely to exhibit emergence behavior than others, see §3.2. Instead of entropies, we believe that the regularity of the optimal response function plays a key role in the emergence of these abilities across different tasks. Our main argument is that LLMs tend to learn more effectively when the local derivative is small, and sacrifice learning tasks where the local derivative is large for better performance elsewhere. Under reasonable assumptions, our main theorem (Theorem 2.5 verifies mathematically that this is indeed a favorable response policy. This theoretical interpretation is then

supported by experiments in §3, both with a ResNet based toy model where derivatives of the target function are explicitly kept track of, as well as with LLM's on arithmetic tasks where methods are avilable to estimate the size of the derivatives.

Methods for improving performance of LLM model commonly includes fine-tuning (Devlin et al., 2019; Brown et al., 2020; Raffel et al., 2020; Liu et al., 2020; Sun et al., 2019; Hu et al., 2022; Houlsby et al., 2019) and Chain-of-thought (CoT) (Wei et al., 2022b; Wang et al., 2023; Gao et al., 2023; Yao et al., 2023; Zhang et al., 2023; 2024). From our perspective, the reason behind the success of these two method in improving the accuracy of models is that they both decrease the norm of local derivatives of the response function. More discussions are included in §4 to address the links between these methods and our theory.

## 2 STATEMENT OF MAIN RESULTS

### 2.1 PRELIMINARIES

**Settings.** An LLM is a neural network that provides a function $f : \mathcal{X} \to \mathcal{D}(\mathcal{Y})$ where $\mathcal{X}$ is the space of tokenized prompts, and $\mathcal{Y}$ is the space of tokenized answers, and $\mathcal{D}(\mathcal{Y})$ is the space of probability measure on $\mathcal{Y}$. We view $\mathcal{X}$ as a bounded subset inside an ambient space $\mathbb{R}^{d_{\mathcal{X}}}$. The natural distribution of token's in $\mathcal{X}$ is characterized by a probability measure $\mu$. The neural network is trained to minimize a loss functional $L(f)$.

The optimal loss of an LLM of scale $N$ trained on $D$ random samples drawn from $(X, \mu)$ is denoted by $L(N, D)$. A standard decomposition in machine learning literature partitions it into three pieces

$$L(N, D) = L(f^*) + \big(L(f_N) - L(f^*)\big) + \big(L(f_{N,D}) - L(f_N)\big). \tag{1}$$

Here $f^*$ is the minimizer of $L$, $f^* = \arg\min_f L(f)$. $f_N$ is the minimizer within a given family of models whose number of parameters are bounded by $N$. $f_{N,D}$ is the "single epoch empirical risk" minimizer over the dataset of size $D$.

A popularly accepted scaling law for LLM is:

**Assumption 2.1.** *[(Hoffmann et al., 2024), eq(2)] The parametric loss function of an LLM is*

$$L(N, D) = E + AN^{-\alpha} + BD^{-\beta},$$

*where $N$ is number of parameters and $D$ is size of training data.*

Here $E$ is the intrinsic optimal loss that measures the natural uncertainty of the answer. The term $AN^{-\alpha}$ corresponds to the second term in (1) and measures the ability for a model $f_N$ with parameters $\theta \in \mathbb{R}^N$ to approximate an arbitrary function $f(x) = y$ for $x \in \mathcal{X}, y \in \mathcal{Y}$ are respectively prompt and answer. The theoretical support behind this term is the belief that the optimal error between $f_N$ and $f$ is $O(N^{-\alpha})$. Quoting from (Hoffmann et al., 2024):

*" In the decomposition (9), the second term depends entirely on the number of parameters $N$ that defines the size of the functional approximation space. On the set of two-layer neural networks, it is expected to be proportional to $\frac{1}{N^{\frac{1}{2}}}$ ((Siegel & Xu, 2020))... Empirically, we find ... that $L(N, D) = E + \frac{A}{N^{0.34}} + \frac{B}{D^{0.28}}$. "*

Following the paper (Siegel & Xu, 2020) cited by (Hoffmann et al., 2024) as mathematical basis, the constant $A$ is related to the Baron norm of function $f$.

**Definition 2.2.** *The Barron norm at order $s$ of a function $f$ on $\mathbb{R}^d$ is*

$$\|f\|_{\mathcal{B}^s} = \int_{\mathbb{R}^d} (1 + |\omega|^s)|\hat{f}(\omega)|\mathrm{d}\omega.$$

**Definition 2.3.** *The Sobolev norm at order $s$ of a function $f$ on $\mathbb{R}^d$ is*

$$\|f\|_{H^s} = \Big( \int_{\mathbb{R}^d} (1 + |\omega|^{2s})|\hat{f}(\omega)|^{2s}\mathrm{d}\omega \Big)^{\frac{1}{2}}$$

.

It was proved in (Siegel & Xu, 2020) that, for 2-layer neural networks, under the assumption that the ground truth function $f$ satisfies

$$\|f\|_{\mathcal{B}^s} < \infty, \tag{2}$$

then

$$\inf_\theta \|f - f_N\|_{H^s} \ll N^{-\frac{1}{2}} \|f\|_{\mathcal{B}^{s+1}}. \tag{3}$$

Note that when $s = 0$, $\|\cdot\|_{H^0}$ is just $\|\cdot\|_{L^2(\mathcal{X})}$ with respect to the Lebesgue measure on $\mathcal{X}$, and by (3) it is controlled by $\|f\|_{\mathcal{B}^1}$. A similar inequality also holds for an arbitrary unspecified measure $\mu$: it follows from combining (E et al., 2022)[Theorem 1] and (Wu, 2023)[Theorem 1.4] that,

$$\inf_\theta \|f - f_N\|_{L^1(\mu)} < \inf_\theta \|f - f_N\|_{L^2(\mu)} \ll N^{-\frac{1}{2}} \|f\|_{\mathcal{B}^2}. \tag{4}$$

Here the first inequality trivially holds because $\mu$ is a probability measure.

If the neural network is evaluated in MSE loss, then (4) controls the approximation loss $L(f_N) - L(f^*)$. Most LLM architectures uses cross entropy as the loss function, in this case, after viewing the final softmax layer and another log layer as parts of the network, the loss function can be regarded as a $L^1$-loss as $\sum p_i \log \frac{p_i}{q_i} = \mathbb{E}(\log p_i - \log q_i) \leq \mathbb{E}|\log p_i - \log q_i|$. So the loss function is also controlled in this case as long as (4) holds.

In light of the reasoning in (Hoffmann et al., 2024), the inequality (4) and the discussion above, we make the following refined assumption on the second term of the scaling law (2.1):

**Assumption 2.4.** *The architecture of LLM satisfies: there exists $A_0, \alpha > 0$ and $s \geq 1$ such that for all $\|f\|$ with $\|f\|_{\mathcal{B}^s} < \infty$,*

$$L(f_N) - L(f) \ll A_0 N^{-\alpha} \|f\|_{\mathcal{B}^s}. \tag{5}$$

The adoption of the Baron norm $\|\cdot\|_{\mathcal{B}^s}$ in (Siegel & Xu, 2020) and (E et al., 2022) successfully interprets the absence of curse of dimensionality (CoD) in practical training of large neural nets as the exponent $N^{-\frac{1}{2}}$ is better than the $N^{-\frac{1}{d}}$ in CoD. However, the Baron norm is known to dominate Sobolev norms of similar orders (see e.g. (Siegel & Xu, 2020)[Lemma2]) but not equivalent to them. While a function may simultaneously have bounded Sobolev norm and unbounded pointwise derivatives, this is not true for Barron norms. In the paper (Barron, 1993) that introduced Barron norms, it was noted that in order to have bounded $\|f\|_{\mathcal{B}^1}$ (and hence all $\|f\|_{\mathcal{B}^s}$ $s \geq 1$): *"it is necessary (but not sufficient) that all first order partial derivatives be bounded."*

Recall that, in the setting of natural language models, $f^* : \mathcal{X} \to \mathcal{P}(\mathcal{Y})$ is the optimal probabilistic mapping from prompt to response. Given the complexity of inquiries behind possible prompts, the responder may need to process an arbitrarily large amount of information in order to give an accurate answer. For instance, the prompt "Could you explain the mechanism of $X_0$ and give an example?" where $X_0$ is a scientific or technical term. The optimal responses are then highly sensitive to the single token $X_0$. In other words, the derivative $Df^*$ has very large size for this particular prompt. The tail distribution of such hard prompts makes the size of derivative $|Df^*|$ unbounded on $\mathcal{X}$. By the remark above, $\|f^*\|_{\mathcal{B}^s}$ becomes unbounded in this case, making Assumption 2.4 inapplicable. To digest this obstacle, we interpret emergence as a natural choice of LLM when handling prompts with low regularity (large derivatives).

## 2.2 MAIN THEOREM

**Main Theorem (heuristic statement)** *Instead of predicting $f^*$, the model will fit $f_N$ to approximate a smoothified function $\mathcal{S}_\epsilon f^*$ with bounded $\|\cdot\|_{\mathcal{B}^s}$ norm. The substitute $\mathcal{S}_\epsilon f^*$ itself is a perturbation of $f^*$ of the averaged form*

$$\mathcal{S}_\epsilon f = \mathbb{E}_{\triangle x \sim \mathcal{N}(0,I)} f(x + \epsilon \triangle x), \tag{6}$$

*and deviates from $f^*$ substantially only near where $|Df^*| \gg \frac{1}{\epsilon}$. The granularity $\epsilon$ of the approximation is a function of $N$ and $\lim_{N \to \infty} \epsilon = 0$.*

Following (Siegel & Xu, 2020), we define a Gaussian mollifier of scale $\epsilon$ on $\mathbb{R}^{d_\mathcal{X}}$ by $\eta_\epsilon(x) := \mathcal{N}(0, \epsilon I, \mathbb{R}^{d_\mathcal{X}}) = \frac{1}{(\pi^{\frac{1}{2}} \epsilon)^{d_\mathcal{X}}} e^{-\frac{\|x\|^2}{2\epsilon^2}}$ on $\mathbb{R}^{d_\mathcal{X}}$ and a corresponding smoothing operator

$$\mathcal{S}_\epsilon f := f \star \eta_\epsilon, \tag{7}$$

where $\star$ denotes the convolution. This is equivalent to the definition (6). As $\epsilon \to 0$, $\eta_\epsilon$ converges to the Dirac mass and $\mathcal{S}_\epsilon f(x) \to f(x)$ for all Schwartz functions.

We now state the mathematical statement of the main theorem.

**Theorem 2.5.** *(Main Theorem) Under assumptions 2.4, if $|Df^*|$ is unbounded on $\mathcal{X}$, then there is an optimal value $\epsilon = \epsilon(N) > 0$, such that:*

1. *Instead of the upper bound (5) which yields an infinite value, the LLM will obey the scaling law*

$$L(f_N) - L(f^*) \leq A_0 N^{-\alpha} \|\mathcal{S}_\epsilon f^*\|_{\mathcal{B}^s} + B_0 \|\mathcal{S}_\epsilon f^* - f^*\|_{L^1(\mu)}; \tag{8}$$

2. $\epsilon \to 0$ *as* $N \to \infty$;

3. *For any fixed $\delta > 0$, there is $K > 0$ such that $|\mathcal{S}_\epsilon f^*(x) - f^*(x)| \leq \left( \sup_{|z-x|<K\epsilon} |Df^*(z)| \right)\epsilon + \delta$.*

The main takeaway of the main theorem is that, in view of a prescribed precision standard $\delta$, the model would give up predicting $f^*$ for input $x \in \mathcal{X}$ when $\left( \sup_{|z-x|<K\epsilon} |Df^*(z)| \right)K\epsilon$ is large, and predict an averaged value of $f^*$ near $x$. The optimal $\epsilon(N)$ is characterized by the trade-off between two terms on the right hand side of (8). A priori, it could be $\infty$. When this happens, it means the scale $N$ is too small compared to the derivatives of $f^*$, and the model wouldn't try to fit any region of $\mathcal{X}$. However $\epsilon$ is finite for large $N$.

The proof of Theorem 2.5 is postponed to Appendix A.1. See also Figure 11 in the appendix for an illustration of the theorem.

## 3 EXPERIMENTS

We implemented several experiments to verify the interpretation offered by Theorem 2.5. As the theorem is not specific to transformer-based LLM models, we first verify it in a toy model setting (3.1) of fitting a trigonometric function with a family of ResNet's whose scales get multiplied by up to $2^{15} = 32768$ times from the smallest to the largest.

We then test arithmetic tasks in §3.2 - §3.4) with few-shots prompt using the `Qwen1.5-Chat` family of models (Qwen Team, 2024) . This family is chosen as it has the widest multiplicative span in terms of scale among commonly available models for our comparison purposes, ranging from 0.5B to 110B, a 220-fold increase. All experiments were implemented with the `int4` quantization of models in order to fit the largest model (110B) into an Nvidia A100 GPU while maintaining fair comparison across different scales.

For the arithmetic tasks, we test model performance, measure in accuracy and average error, on individual digits. The results on accuracy are included in this section and those on average error can be found in Appendix C. Because (i) each prompt correspond to a unique answer, i.e. the optimal answer policy has 0 cross entropy; (ii) the response for each digits ranges from 0 to 9, i.e. the cross entropy loss upper bound is always $\log 10$ via uniform random choices, different subtasks of predicting a single digit as part of an arithmetic question (e.g. the 5-th digit in the product of two 4-digit integers) are expected to have the same complexity from the viewpoint of the cross entropy loss function used by the LLM. Nevertheless, our experiments show that the difficulty levels and emergence patterns vary greatly among such subtasks, which is well explained by the regularity of the local derivative $Df^*$ as foreseen by Theorem 2.5.

### 3.1 TOY MODEL: RESNET FOR PREDICTING FUNCTIONS WITH VARYING REGULARITY

The analysis above is not limited to LLM. The following analysis shows the emergence phenomenon when learning a complicated mathematical function. Actually, since $f^* = f$ and its derivative is explicit in this synthetic dataset, the emergence mechanism proposed by the main theorem is better observed in this toy model.

We generated uniformly $2^{18}$ training data points $x \in [0,1]^{16}$ and use a ResNet architecture to learn the function $f : [0,1]^{16} \to \mathbb{R}^2$ given by

$$f(x) = \big( \sin \frac{512}{(\sum_i x_i)^2}, \cos \frac{512}{(\sum_i x_i)^2} \big),$$

which is designed to be a highly irregular function since $|Df(x)| = \frac{2 \cdot \sqrt{16} \cdot 512}{(\sum_i x_i)^3}$ is very large whenever $\sum_i x_i$ is small. With MSE loss, $f^* = f$. We trained 6 different scales $\textbf{ResNet}_k, k = 0, \cdots, 5$ where $\textbf{ResNet}_k$ has $2^k$ residual layers and hidden dimension is $2^{k+4}$ in each layer, hence the number of parameters in $\textbf{ResNet}_k$ is approximately $N(k) \approx 2^{3k+8}$. We then test the data on a test dataset of size $2^{12}$. For each $k$, 8 independent instances of the experiment were ran. We then aggregate results from all instance, bin data points from the test dataset according to the size of $|Df(x)|$ and plot the average prediction error against the average of $|Df(x)|$ inside each bin in Figure 1. One can see that for difficult inputs (large $|Df(x)|$), the model refuses to learn and the error is oscillating around a constant level 1 until the scale of the model reaches a threshold that increases with $|Df(x)|$.

Indeed, in a small neighborhood of $x$ with large $|Df(x)|$, $f(x)$ is a fast rotating point in the unit circle with argument $\frac{512}{(\sum x_i)^2}$. One can easily show the target value $f(x)$ is nearly equidistributed along the united circle and its average inside the neighborhood would be $(0,0)$. That is, given $\epsilon$, $S_\epsilon f \to (0,0)$ as $|Df(x)| \to \infty$. Our main theorem predicts that neural networks models will output $(0,0)$ predictions until their scales $N$ reach a threshold. Experiment results is consistent with this prediction. In Figure 2, it is clear that each given model has two thresholds $D_0(N) < D_1(N)$, both increasing in terms of the scale $N$, for $|Df(x)|$. For $|Df(x)| < D_0$, the model manages to make a prediction inside the unit circle, this is the range where $S_\epsilon f \approx f$. For $|Df(x)| > D_1$, the model is incapable of producing an informative prediction and outputs the average value $S_\epsilon f \approx (0,0)$ instead. These thresholds increase in $N$ because $\epsilon = \epsilon(N) \to 0$.

**Remark 3.1.** *In both Figures 1 & 2, as the model scale $N$ increases, the learning patterns, in particular $D_0(N)$ and $D_1(N)$, translate towards the right parallelly. In order to support Theorem 2.5, it is this pattern that we hope to observed in LLM's.*

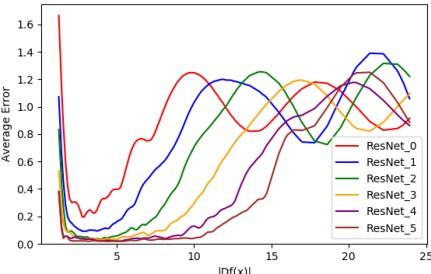

Figure 1: Average error vs $|Df(x)|$, ResNet Experiment

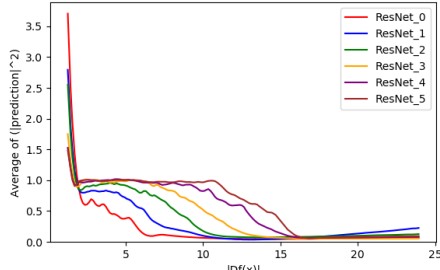

Figure 2: Average of $|\text{prediction}|^2$ vs $|Df(x)|$, ResNet Experiment

## 3.2 MULTIPLICATION OF TWO INTEGERS

Arithmetic multiplication (and other arithmetic tasks) is a well-known difficult obstacles for LLM's (Dziri et al., 2023; Qian et al., 2023). Several recent works (Shen et al., 2023; Yang et al., 2023; Lee et al., 2024) described various fine-tuning methods for overcoming it. This experiment aims to demonstrate that the difficulty is tied to size of derivatives in the token space.

We take two random integers of $d$ digits, and prompt `Qwen1.5-Chat` models to multiply them together. The value of $d$ are either $4, 6$, or $8$. We then compare the response to the correct answer, retrieving the accuracy rate individually for each digit in the answer over 128 random instances with different prompt-answer pairs.

Before continuing, we reason that the problem would be most difficult for a model over the digits in the middle of the product, as the computation of these digits would have larger norms of tokenized derivatives.

In this particular setting, we are refined to a subset of tokens $(x, x') \in \Lambda^d \times \Lambda^d \subset \mathcal{X}$ where $\Lambda = \{\text{'0','1',}\cdots\text{,'9'}\}$ is the set of symbolic tokens representing the digits, embedded as a subset in the ambient Euclidean space of all individual tokens. While we cannot know exactly how $\Lambda$ is embedded, we will assume that it keeps the metric space structure of the set $\{0, 1, \cdots, 9\}$ of numerical values. Hence, by abusing notation, we shall simply identify $\Lambda$ as $\{0, 1, \cdots, 9\} \subseteq R$. Thus we will view $\mathcal{X}$ as as subset in $\mathbb{R}^{2d}$. As the answer of multiplication is unique, the fitting target on $\Lambda^{2d}$ is the restriction $f^*|_{\Lambda^{2d}} : \Lambda^{2d} \rightarrow \Lambda^{2d} \subset \mathbb{R}^{2d}$ (instead of to the space of probability measure on $\Lambda^{2d}$. The dimension is $2d$ because the product between two $d$-digits numbers have either $2d$ or $2d - 1$ digits, and we will always consider it as a sequence of length $2d$ by allowing the leftmost digit to be $0$. Denote the subquestion of determining the $k$-th digit of the product from the left as $f^*_{k,d}(x, x') = y_k$, which is a $\Lambda$-valued function.

Because $\Lambda$ is a discrete space, it is impossible to compute its derivative, we also define $\tilde{x}_j = \overline{x_j.x_{j+1}\cdots x_d}$ and similarly $\tilde{x}'_j$, $\tilde{y}_j$. This allows to define and $\mathbb{R}$-value function $\tilde{f}^*_{k,d}(x, x') = \tilde{y}_k$. Which is an approximation of $f^*_{k,d}$ and $\lfloor \tilde{f}^*_{k,d}(x, x') \rceil = f^*_{k,d}(x, x')$. We will satisfy with this approximation and estimate the derivatives of $\tilde{f}^*_{k,d}$.

**Lemma 3.2.** *For the multiplication experiment, the $L^2$ norm of the gradient vector $D\tilde{f}^*_{k,d}$ satisfies*

$$\|D\tilde{f}^*_{k,d}\|_{L^2(\Lambda^{2d})} \approx \begin{cases} \sqrt{\frac{2}{3}}, & k = 1 \\ \sqrt{\frac{100(k-1)}{3}}, & 2 \leq k \leq d; \\ \sqrt{\frac{100(2d-k)}{3}}, & d+1 \leq k \leq 2d. \end{cases},$$

*where the $L^2$ norm is taken over uniformly drawn $d$-digit integers $x \in \Lambda^d$, $x' \in \Lambda^d$.*

The proof of the lemma is delayed to Appendix A.2.

Based on Theorem 2.5, the ability of learning the function $f^*_{k,d}$ emerges around a scale $N$ that increases with the expected norm of $|f^*_{k,d}|$ on $\Lambda^d \times \Lambda^d$. Therefore, combining Theorem 2.5 and Lemma 15 leads to the follow predictions:

(i) Learning of the 1st digit from the left in the product is much easier than other digits and its emergence should occur at a much smaller model scale, because the expected norm of $|Df^*_1|$ is far smaller than other $|Df^*_{k,d}|$'s.

(ii) Emergence should happen at smaller model scales for the digits near both ends of the product. The pattern of the the emergence should be roughly symmetric between the $k$-th and the $(2d - k + 2)$-th digits for $2 \leq h \leq d$. (In particular, the 2nd digit and the last digit are supposed to be symmetric in behavior.)

(iii) Given $k$, the emergence pattern should be roughly the same across different values of $d \geq k$ for the $k$-th digit both from the left, as well as for the $k$-th digit from the right.

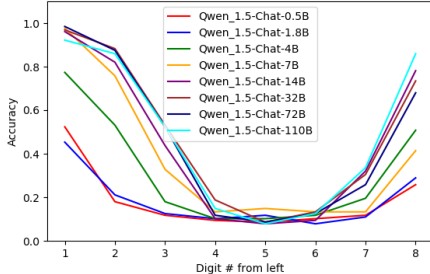
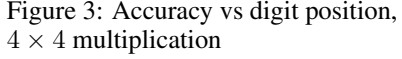
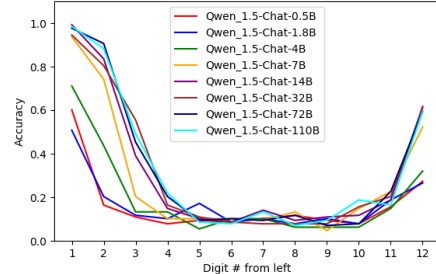

Figure 3: Accuracy vs digit position, $4 \times 4$ multiplication

Figure 4: Accuracy vs digit position, $6 \times 6$ multiplication

Experiment results, presented in Figures 3-5, in general match the above predictions well. Despite of occasional noisy effects that plots cross each other when larger models perform worse on a digit, they

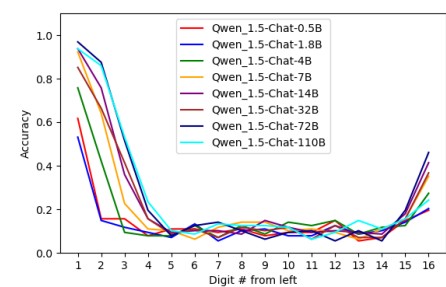

Figure 5: Accuracy vs digit position, $8 \times 8$ multiplication

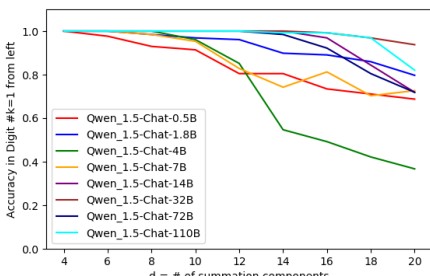

Figure 6: Accuracy vs # summands, digit # $k = 1$, summation experiment

display the desired trend of shifting together towards the center as model size increases. Moreover, the following properties are observed, (i) Accuracy is high for digit #1, even with small scale models. (ii) The emergence pattern, i.e. the critical digit position where the accuracy reaches its minimum, is symmetric, centered at the $(d+1)$-th digit as predicted (digits #5,#7,#9 respectively for $d = 4, 6, 8$). In terms of the actual accuracy value, the results are still quite symmetric around the center at digit #$d + 1$, observed better for $d = 4$ but less so for $d = 6, 8$ with digits on the left side outperforming those on the right. We hypothesize that this is because our theoretical analysis above treats all $f_{k,d}^*$'s as independent task, while in actual inference they are processed sequentially, taking outputs from earlier digits as new inputs and thus adding noise if those outputs are not accurate. (iii) The patterns for the beginning digits counting from either end are approximately the same when $d$ varies. The critical emergence scales occur at similar scale-position pairs for different $d$'s. For actual accuracy value, this phenomenon is better observed near the left end, probably due to the some reason of sequential inference suggested earlier for property (ii).

### 3.3 SUMMATION OF SINGLE-DIGIT INTEGERS

In this experiment, we take $d$ random single digit numbers from $\Lambda = \{0, \cdots, 9\}$, and prompt `Qwen1.5-Chat` models of different sizes to compute their sum. The value of $d$ ranges over all even integers from 4 to 20. Like in §3.2, for each $s$, we sample 128 random prompts and calculate the accuracy in each digit. Because the correct answer can have up to 3 digits for the largest $s$, in order to better compare the statistics for all choices of $d$ we extend answers to 3 digits by adding 0's to the left.

Following the same analysis framework from §3.2 under the assumption that the tokenization of $\Lambda$ respects its metric space properties, the question of summing $d$ numbers is fitting the function $f^* : \Lambda^d \to \Lambda^3$ that sends $x = (x_1, \cdots, x_d)$ to $y = (y_1, \cdots, y_3)$ such that $\overline{y_1 y_2 y_3} = \sum_{i=1}^d x_i$. As before, let $f_{k,d}^*(x) = y_k$ be the map predicting the $d$-th digit from right. Again, since $\tilde{f}_{k,d}^*$ jumps between discrete values, we approximate it with $\tilde{f}_{k,d}^*$ which sends $x$ to

$$\tilde{y}_3 = y_3, \tilde{y}_2 = \overline{y_2 . y_3}, \text{ or } \tilde{y}_1 = \overline{y_1 . y_2 y_3} \tag{9}$$

respectively for $k = 3, 2, 1$.

**Lemma 3.3.** *For the summation experiment,* $|Df^*_{k,d}| = 10^{k-3}\sqrt{d}$.

*Proof.* Because $\tilde{y}_k = (10^{k-3}\sum_{i=1}^d x_i) \bmod 10$, $\frac{\partial \tilde{y}_k}{\partial x_i} = 10^{k-3}$ for all $i$. Since $Df^*_{k,d} = (\frac{\partial \tilde{y}_k}{\partial x_1}, \cdots, \frac{\partial \tilde{y}_k}{\partial x_d})$, The lemma immediately follows. $\square$

The predictions from combining the estimate in Lemma 3.3 and Theorem 2.5 are:

(i) As the number of components $d$, as well as the output digit position $k$ increases, learning become hard and there is a threshold $N_1 = N_1(k, d)$ at which emergence happens, i.e. model refuses to learn for $N < N_1$. $N_1$ is increasing in both $k$ and $d$.

(ii) The emergence is far more sensitive to $k$ instead of to $d$ as $|Df^*_{k,d}|$ varies exponentially in $k$ but as a square root in $d$.

Results from the experiments, shown as Figures 6-6, well exhibits these expected trends despite of a few noisy instances where curves cross each other or swap orders. When $d$ increases, the curves shift towards the right together. Moreover, the emergence pattern (where the accuracy curves reach the bottom) is very sensitive on $k$ as predicted, the accuracy figures for $k = 1, 2, 3$ are very different, in view of the $(N, d)$ pairs where the plots touch the bottom (critical phase for emergence). When $k$ decreases from 3 to 2, for each fixed $N$, the critical $d$ increases a lot, actually for half of $N$'s it moves out of the tested range $[4, 20]$. When $k$ further decreases to 1, the critical $d$ for all $N$'s moves out of the range $[4, 20]$, likely far beyond 20 telling from the plots.

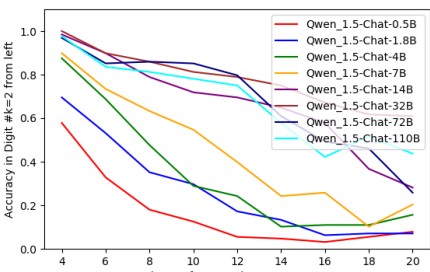

Figure 7: Accuracy vs # summands, digit # $k = 2$, summation experiment

Figure 8: Accuracy vs # summands, digit # $k = 3$, summation experiment

### 3.4 ADDITIVE REASONING

In this experiment, we ask `Qwen1.5-Chat` models to work on integer summation tasks very similar to those from §3.3 in nature. However, these tasks are formulated in a verbal reasoning setting. To be precise, we input, after a few-shots paragraph, a prompts questions of the form *"Gary is 173cm tall. Grace is as tall as Gary. Jack is 13cm shorter than Grace. Tina is 28cm taller than Jack. How tall is Tina?"* We tested question with $d$ steps for $k = 1, \cdots, 6$. The example above involves 4 characters and is counted as a 3-step question. All involved characters are assigned an integer height value in cm randomly chosen between 150 and 199.

A main difference with the previous experiment, besides having summands from different ranges, is that the LLM now is supposed to decode the meaning of each individual step *"Jack is 13cm shorter than Grace."*, which identifies Jack's height as function of Tina's. Assuming this mechanism, we view the optimal response function $f^*$, restricted from $\mathcal{X}$ to the family of prompts in this experiment, as a map from $\Theta \times \Lambda^d$ to $\Theta$ where $\Theta = \{150, \cdots, 199\}$ and $\Lambda = \{$"shorter", "as tall as", "taller"$\} \times \{0, \cdots, 49\}$. Instead of a direct arithmetic summation $x, \{(\sigma_i, \delta_i)\}_{i=1}^d) \rightarrow x + \sum_i^d \sigma_i \delta_i$ where $\sigma_i \in \{-1, 0, 1\}$, $f^*|_{\Theta \times \Lambda^d}$ should be viewed as a $k$-fold composition of a 1-step function $g : \Theta \times \Lambda \rightarrow \Theta$: $f^*(x, \lambda_1, \cdots, \lambda_d) = g(\cdots (g(g(x, \lambda_1), \lambda_2)) \cdots, \lambda_d)$. Because the inference is now run individually for each step, we expect the derivative norm of the one step function $g$ to be approximately a constant $C > 1$ on average due to the repeated composition, and thus that

$$\|Df^*|_{\Theta \times \Lambda^d}\| \asymp C^d, \tag{10}$$

(i.e. has positive Lyapunov exponent in control-theoretic terms.)

As in (9) denote by $f_*^{k,d} : \Gamma \times \Theta^d \to \Lambda = \{0, \cdots, 9\}$ the function that maps to the $k$-th digit $y_k$ in the correct answer $\overline{y_1 y_2 y_3}$, and approximate it by an $\mathbb{R}$-valued map $\tilde{f}_*^{k,d}$ that maps to $\tilde{y}_k^*$. Assuming (10), as in Lemma 3.3 we deduce that

$$\|D\tilde{f}_{k,d}^*\| \asymp 10^{k-3}C^d. \tag{11}$$

We expect similar emergence patterns to those in §3.3 as $k$ and $d$ increases, with the difference that in the current experiment the emergence pattern is sensitive to both $k$ and $d$ as $\|D\tilde{f}_{k,d}^*\|$ increases exponentially with respect to both of them. In particular, we expect that emergence quickly gets seriously obstructed even at smaller $N$'s (compared to §3.3) as the step number $d$ grows.

We include the accuracy results for digits at positions $k = 2, 3$ in Figures 9-10. The digit at $k = 1$ is ignored as its value is always 1 for our data distribution and is not an interesting learning problem. The curves again shift towards the right as $N$ like in previous experiments. The main observation we want to make is that when $k = 3$, the step numbers $d = 5, 6$ shows no sign of emergence at all in Figure 10 even with the largest model size. The contrast to Figure 8, where emergence occurs at up to $d = 14$ summands, supports our prediction that emergence quickly becomes difficult when $d$ increases.

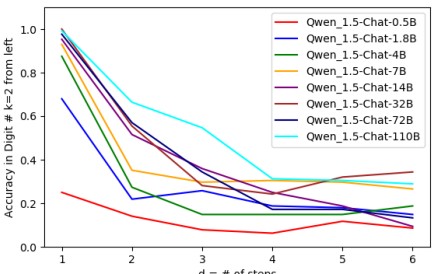

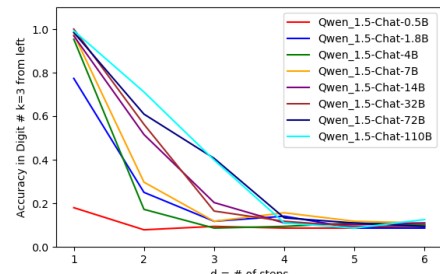

Figure 9: Accuracy vs # steps, digit # $k = 2$, heights experiment

Figure 10: Accuracy vs # steps, digit # $k = 3$, heights experiment

## 4 RELATIONS TO FINE-TUNING AND CHAIN-OF-THOUGHT

Theorem 2.5 provides theoretical ground to explain two methods of performance enhancement for LLM's: fine-tuning and chain-of-thoughts.

### 4.1 INTERPRETATION OF FINE-TUNING

Fine-tuning is an important technique for improving the performance of LLM in specific domains. There is extensive discussion of fine-tuning methods in the literature (Devlin et al., 2019; Brown et al., 2020; Raffel et al., 2020; Liu et al., 2020; Sun et al., 2019; Hu et al., 2022; Houlsby et al., 2019). The approach involves taking an LLM and further train it on a new dataset that represents a special family of the task to achieve better performance on that family, sometimes leading to emergence of abilities that were not present in the original LLM trained with common data. One such example is Yang et al. (2023), where emergence of of arithmetic operation abilities are observed on moderately sized models ($\leq$6B) trained with extensive data (up to 50M training samples) specific to arithmetic tasks. Following the argument in the proof of Theorem 2.5, this phenomenon could be explained by approximation accuracy in terms of regularity.

Providing enough data to train exclusively on special tasks focuses the learning onto a small subset $\mathcal{X}_0 \subset \mathcal{X}$. For difficult tasks such as multiplication, the optimal answering policy $f^*$ may have very low regularity on $\mathcal{X}$ as demonstrated in §3.2. On the other hand $\mathcal{X}$ is tiny in size as as subset of the ambient dataset $\mathcal{X}$ which broadly consists of all available natural language paragraphs.

Recall that $\mathcal{X} \subset \mathbb{R}^{d_{\mathcal{X}}}$ is a compact set because of the finiteness of the token sets, and we can think of it as $[0, 1]^{d_{\mathcal{X}}}$ without loss of generality. In fine-tuning, the training is zoomed into a small subdomain

$\mathcal{X}_0$, which we will assume to be of diameter $r$ which is far smaller than 1. Because the inputs are on a smaller scale, after automatically adjusting internal weights, the model can renormalize this domain to unit size and train. Making the simplistic assumption that the renormalizing map has the form $x \to \frac{x}{r} + b$, The renormalized map $f_{\text{new}}^*$ will have the form $f_{\text{new}}^*(x) = f^*(r(x-b))$, which scales the norm of $|Df^*|$ by $r$.

By the discussion in §2.2, for each given scale $N$ there is an optimal scale parameter $\epsilon(N)$ such that the critical regime for emergence is near $|Df^*| \asymp \frac{1}{\epsilon(N)}$. Thus depending on the original size of $\left|Df^*|_{\mathcal{X}_0}\right|_{L^\infty}$, when the diameter $r$ of $\mathcal{X}_0$ is sufficiently small (of order $O\left(\frac{1}{\epsilon(N)\left|Df^*|_{\mathcal{X}_0}\right|_{L^\infty}}\right)$), or in other words the training data is sufficiently specific compared to the irregularity of the task, the ability for the prescribed tasks would emerge.

### 4.2 Interpretation of Chain-of-Thought prompting

Our proposed theory also explains naturally how Chain-of-Thought (CoT) improves reasoning on complex tasks. Chain-of-Thought can be viewed as the decomposition of a complex function $f$ into multiple steps $f = f_1 \circ f_2 \circ \cdots \circ f_d$. We refer interested readers to the foundational works on Chain-of-Thought (CoT), such as (Wei et al., 2022b; Wang et al., 2023; Gao et al., 2023; Yao et al., 2023; Zhang et al., 2023; 2024). By the chain rule, the derivative of $f$ becomes the product of the derivative of the individual $f_i$, and its norm is approximately the product of the norm of steps. By Theorem 2.5, given model scale $N$, prompts representing problems whose derivative norm are beyond a certain threshold $D_1(N)$ will be given up by the model. If $\|Df\| > D_1(N)$, decomposition would allow the derivative norm of each component to get below this threshold and be individually learned. The experiment in §3.4 is a good example of this: the accuracy for answering the 1-step question is much higher than for multiple step ones. With proper decomposition into intermediate prompts, the number of needed steps in CoT is expected grow logarithmically with the expected norm of $|Df^*|$.

## 5 Conclusions, limitations and future directions

We propose an underlying mechanism for the emergence ability of large language models. This aligns with the well known observation that use of few-shot prompt engineering and Chain-of-Thought (CoT) reasoning can enhance model performance in downstream tasks, as well as shed lights on why tasks like multiplication of multi-digit integers are difficult for LLMs. We demonstrate the link between the magnitude of derivatives and emergence abilitilities in various examples, and provide evidences from both theoretical and experimental perspectives.

One direction left out in this work is to explore possible ways to reduce the magnitude of derivatives. As a target function is unknown, specific methods to detect where its derivatives are large, and to reduce the magnitude is easily available. In this paper, we examined specific tasks like summation, multiplication, and composition of subtasks, where special task structures allowed us to estimate the magnitude of derivatives. This may shed light on how to measure derivatives of more general questions. While rescaling is the a natural way to reduce the magnitude of derivatives, it would be interesting to extend the study to other methods. For instance, we believe variations of convolution with mollifiers can be a practical method of potential interest. It would also be reasonable to further investigate cut-off operators to improve the regularity of target functions.

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

# A PROOFS OF STATEMENTS

## A.1 PROOF OF THEOREM 2.5

*Proof of Theorem 2.5.* Since $f_N$ is the minimizer among models of scale $N$, we know

$$|L(f_N) - L(f^*)| \leq |L(f_N) - L(S_\epsilon f^*)| + |L(S_\epsilon f^*) - L(f^*)|. \tag{12}$$

We now apply Assumption 2.4. Hence

$$|L(f_N) - L(S_\epsilon f^*)| \ll A_0 N^{-\alpha} \|S_\epsilon f^*\|_{\mathcal{B}^s}. \tag{13}$$

On the other hand, by the discussion before Assumption 2.1,

$$|L(S_\epsilon f^*) - L(f^*)| \leq B_0 \|S_\epsilon f^* - f^*\|_{L^1(\mu)}. \tag{14}$$

*Part 2.* We first show that for sufficiently large $N$, $\epsilon(N)$ is finite and nonzero. Since

$$\|S_\epsilon f^*\|_{\mathcal{B}^s} = \int_{\mathbb{R}^{d_\mathcal{X}}} (1 + |\omega|^s) \cdot |\hat{\eta}_\epsilon| \cdot |\hat{f^*}(\omega)| d\omega,$$

we have

$$\|S_\epsilon f^*\|_{\mathcal{B}^s} \to \|f^*\|_{\mathcal{B}^s} = \infty \quad \text{as} \quad \epsilon \to 0,$$

and

$$\|S_\epsilon f^*\|_{\mathcal{B}^s} \to 0 \quad \text{as} \quad \epsilon \to \infty.$$

On the other hand,

$$\|S_\epsilon f^* - f^*\|_{L^1(\mu)} \to 0 \quad \text{as} \quad \epsilon \to 0.$$

In particular, there exists $\epsilon_0$, such that for any $\epsilon \in (0, \epsilon_0)$,

$$\|S_\epsilon f^* - f^*\|_{L^1(\mu)} < \frac{1}{2} C_0 < \infty.$$

Thus for a very large value $N$,

$$N^{-\alpha} \|S_\epsilon f^*\|_{\mathcal{B}^s} + \|S_\epsilon f^* - f^*\|_{L^1(\mu)}$$

tends to $\infty$ as $\epsilon \to 0$; is $< C_0$ for all $\epsilon \in (\frac{1}{2}\epsilon_0, \epsilon_0)$; and tends to the finite value $C_0$ as $\epsilon \to \infty$. Therefore the optimal bound (minimum value) must be achieved at a finite, non-zero $\epsilon = \epsilon(N) \neq 0$.

From (7) and Definition 2.2,

$$N^{-\alpha}\|S_\epsilon f^*\|_{\mathcal{B}^s} = N^{-\alpha} \int_{\mathbb{R}^{d_\mathcal{X}}} (1 + |\omega|^s) \cdot |\hat{\eta}_\epsilon| \cdot |\hat{f}^*(\omega)| d\omega$$

$$= N^{-\alpha} \int_{\mathbb{R}^{d_\mathcal{X}}} (1 + |\omega|^s) \cdot \frac{1}{(\pi^{\frac{1}{2}})^{d_\mathcal{X}}} e^{-\frac{\|\epsilon\omega\|^2}{2}} \cdot |\hat{f}^*(\omega)| d\omega.$$

This term tends to 0 if $\epsilon$ is bounded and $N \to \infty$. Also, from the Lebesgue dominated convergence theorem, the second term

$$\|S_\epsilon f^* - f^*\|_{L^1(\mu)} \to 0$$

as $\epsilon \to 0$. Thus in order to achieve the optimal value of right hand side of (8), $\epsilon(N) \to 0$ as $N \to 0$.

Finally, to prove part 3 of the statement,

$$|S_\epsilon f^*(x) - f^*(x)|$$

$$= \left| \int \eta_\epsilon(y) f(x - y) dy - \int \eta_\epsilon(y) f(x) dy \right|$$

$$\leq \left| \int_{|y| \leq K\epsilon} \eta_\epsilon(y) |Df(x - t^*y)| \cdot |y| dy \right| + \int_{|y| \geq K\epsilon} \eta_\epsilon(y) |f(x - y) - f(x)| dy$$

$$\leq \left( \sup_{|z-x| \leq K\epsilon} |Df(z)| \right) \cdot (K\epsilon) \cdot \int_{|y| \leq K\epsilon} \eta_\epsilon(y) dy + 2\|f\|_{L^\infty} \int_{|y| \geq K\epsilon} \eta_\epsilon(y) dy$$

$$= p(K) \left( \sup_{|z-x| \leq K\epsilon} |Df(z)| \right) \epsilon + 2(1 - p(K))\|f\|_{L^\infty}$$

$$\leq \left( \sup_{|z-x| \leq K\epsilon} |Df(z)| \right) \epsilon + 2(1 - p(K))\|f\|_{L^\infty}.$$

The last line is derived using $L^1$ dilation.

$$\int_{|y| \leq K\epsilon} \eta_\epsilon(y) dy = \int_{|y| \leq K} \eta(y) dy = p(K),$$

where $p(K) \to 1$ as $K \to \infty$. And apparently $\int_{|y| \leq K\epsilon} \eta_\epsilon(y) dy = 1 - p(K)$. Now, since $\|f\|_{L^\infty}$ is bounded, we derive Part 3 by choosing a fixed $K = K(\delta)$ such that $1 - p(K) \leq \frac{\delta}{2\|f\|_{L^\infty}}$. $\qquad \square$

## A.2 PROOF OF LEMMA 3.2

We now prove the main theorem. Some ideas in the proof are illustrated in Figure 11 below.

*Proof of Lemma 3.2.* Write $x = (x_1, \cdots, x_d)$, $x' = (x_1', \cdots, x_d')$ and $y = (y_1, \cdots, y_{2d})$. Denote by convention $x_j = x_j' = 0$ for all $j > d$. We also define $\tilde{x}_j = \overline{x_j.x_{j+1} \cdots x_d}$ and similarly $\tilde{x}_j'$, $\tilde{y}_j$.

For $1 \leq k \leq 2d$, $y_k = f_{k,d}^*(x, x')$. Observe that $y_k$ only depends on $x_j$ for $j \geq k - d$. In addition, for all $\max(1, k - d) \leq j \leq d$, $x_j$ only affects $y_k$ through its interaction with the $x_l'$'s where $\max(k - j + 1, 1) \leq l \leq d$. Indeed, $y_k = \lfloor \tilde{y}_k \rceil$ where $\tilde{y}_k = ((x_j \tilde{x}_{k-j}' + R) \bmod 10)$ where $R$ is a value determined by terms other than $x_j$. Hence we have $\frac{\partial \tilde{y}_k}{\partial x_j} = \tilde{x}_{k-j}'$. Because that the leading non-zero digit in $\tilde{x}_{k-j+1}'$ is $x_{\max(k-j,1)}'$, we know that $\frac{\partial \tilde{y}_k}{\partial x_j}$ is distributed in the interval $[0, 10 \cdot 10^{-\max(k-j,1)+(k-j)}) = [0, 10^{\min(1, k-j)})$. For uniformly drawn $x$ and $x'$, the distribution in this interval is roughly uniform modulo discretization. Thus

$$\mathbb{E}|\frac{\partial \tilde{y}_k}{\partial x_j}|^2 \approx \mathbb{E}_{t \in [0,1]}(t \cdot 10^{\min(1, k-j)})^2 = \frac{1}{3} \cdot 10^{2\min(1, k-j)}$$

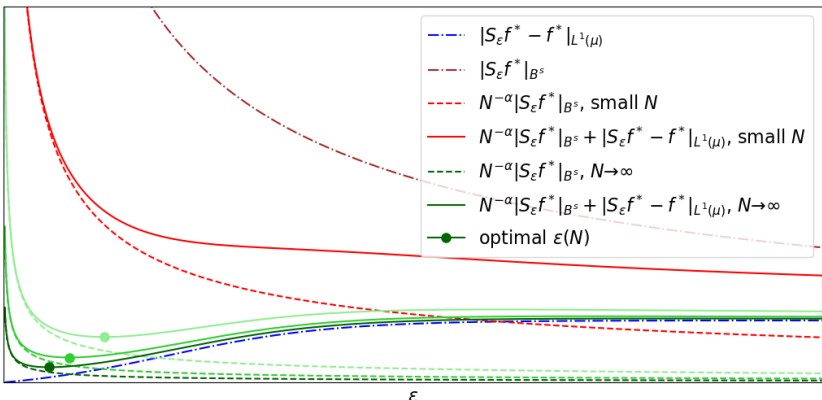

Figure 11: Illustration, Theorem 2.5

Aggregating over all $\max(k-d,1) \leq j \leq d$, it follows that

$$\mathbb{E}\left|\frac{D\tilde{y}_k}{Dx}\right|^2 \approx \frac{1}{3} \sum_{j=\max(k-d,1)}^{d} 10^{2\min(1,(k-j))}. \tag{15}$$

For $j \geq k$, $10^{2\min(1,(k-j))} \leq 1$. Moreover, the sum of all such expressions are at most $\frac{1}{1-10^{-2}} \approx 1$. We will view the part involving such $j$'s as negligible if there is at least one summand that is equal to $10^2$, i.e when $j < k$, in the summation inside (15).

If $k = 1$, then $j \leq k$ for all possible $j$'s and the leading term is at $j = k$, hence $(15) \approx \frac{1}{3}$.

If $2 \leq k \leq 2d$, there is at least one summand where $\max(1, k-d) \leq j \leq d$ and $j < k$, or equivalently the summand is approximately 1. The number of such summand is $k - \max(k-d,1)$, which equals $k-1$ if $2 \leq k \leq d$ and $2d-k$ if $d+1 \leq k \leq 2d$. Thus $E\left|\frac{D\tilde{y}_k}{Dx}\right|^2 \approx \frac{k-1}{3} \cdot 10^2$ in the first case and $\frac{2d-k}{3} \cdot 10^2$ in the latter.

The lemma now follows from the fact that $|D\tilde{f}_{k,d}^*| = \left(\mathbb{E}\left|\frac{D\tilde{y}_k}{Dx}\right|^2 + \mathbb{E}\left|\frac{D\tilde{y}_k}{Dx'}\right|^2\right)^{\frac{1}{2}} = \left(2\mathbb{E}\left|\frac{D\tilde{y}_k}{Dx}\right|^2\right)^{\frac{1}{2}}$, which is because $x$ and $x'$ play symmetric roles. □

## B  SAMPLE PROMPTS IN EXPERIMENTS

1. A sample prompt from the experiment in §3.2, including few-shot instructions :

```
[
    {'role': 'system',
     'content': 'You are a helpful math AI that is good at multiplication.'},
    {'role': 'user', 'content': '3*4'},
    {'role': 'model', 'content': '12'},
    {'role': 'user', 'content': '13*14'},
    {'role': 'model', 'content': '182'},
    {'role': 'user',
     'content': 'What is the final answer of 320970*472234?'}
]
```

2. A sample prompt from the experiment in §3.3, including few-shot instructions :

```
[
```

```
    {'role': 'system',
     'content': 'You are a helpful math AI that is good at summation.'},
    {'role': 'user', 'content': '3+4+5+6'},
    {'role': 'model', 'content': '18'},
    {'role': 'user', 'content': '1+2+4+6'},
    {'role': 'model', 'content': '13'},
    {'role': 'user',
     'content': 'What is the final answer of 5+7+6+0+1+6+1+7'}
]
```

3. A sample prompt from the experiment in §3.4, including few-shot instructions :

```
[
    {'role': 'system',
     'content': 'Answer each question using one integer followed by "cm",
        e.g. "171cm". Examples: \n
        "Jordan is 166cm tall. Grace is 4cm shorter than Jordan. How tall
         is Grace?":"162cm", \n
        "Diana is 157cm tall. Joyce is 13cm taller than Diana. How tall
         is Joyce?":"170cm",\n
        "Lee is 171cm tall. Gary is 3cm shorter than Lee. How tall is
         Gary?":"168cm",\n
        "Howard is 178cm tall. Travis is 1cm taller than Howard.
         Samuel is 6cm taller than Travis. How tall is Samuel?":"185cm",\n
        "Henry is 197cm tall. Alexander is 37cm shorter than Henry. Brenda
         is 9cm shorter than Alexander. How tall is Brenda?":"151cm",\n
        "Thomas is 154cm tall. Linda is 20cm taller than Thomas. John is
         25cm taller than Linda. How tall is John?":"199cm",}'},
    {'role': 'user',
     'content': 'Elizabeth is 154cm tall. Tyler is 27cm taller than Elizabeth.
        Janet is 12cm shorter than Tyler. Laura is 15cm taller than Janet.
        How tall is Laura?'}
]
```

## C   AVERAGE ERRORS FROM EXPERIMENTS IN §3

In the main text, we have only included accuracy results. The corresponding results on average error at individual digit are included below. Similar patterns as in those for accuracy, such as symmetricity, and shifting towards the middle (in §3.2 or towards the right (in §3.3 and §3.4), can be observed in these plots.

### C.1   AVERAGE ERRORS FROM EXPERIMENTS IN §3.2

Figures 12-14 are from experiments in §3.2

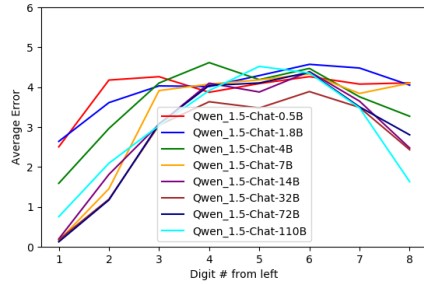
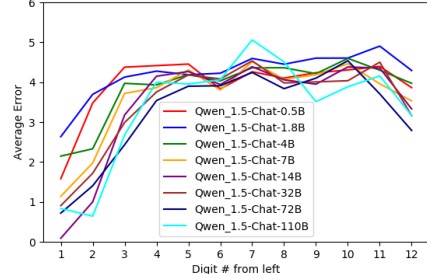

Figure 12: Average error vs digit position, $4 \times 4$ multiplication

Figure 13: Average error vs digit position, $6 \times 6$ multiplication

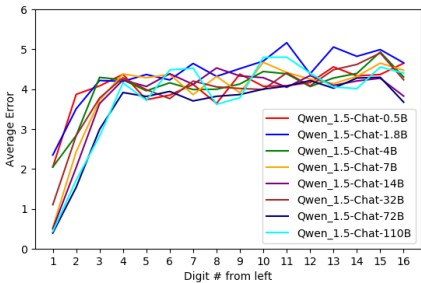

Figure 14: Average error vs digit position, $8 \times 8$ multiplication

## C.2 AVERAGE ERRORS FROM EXPERIMENTS IN §3.3

Figures 15-17 are from experiments in §3.3

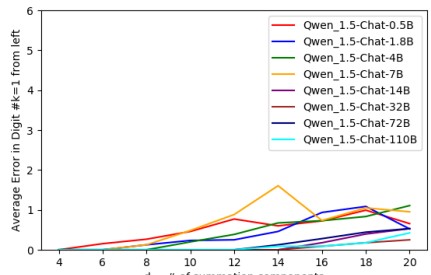

Figure 15: Average error vs # summands, digit #1, summation experiment

Figure 16: Average error vs # summands, digit # $k = 2$, summation experiment

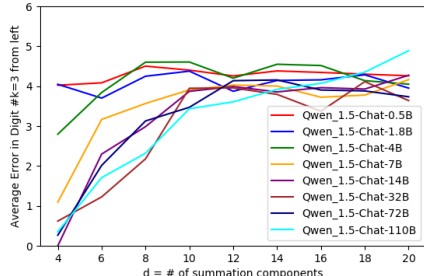

Figure 17: Average error vs # summands, digit # $k = 3$, summation experiment

## C.3 AVERAGE ERRORS FROM EXPERIMENTS IN §3.4

Figures 18-19 are from experiments in §3.4

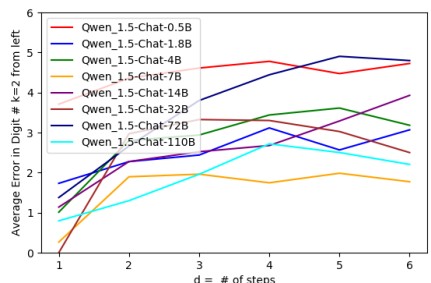

Figure 18: Average error vs # steps, digit # $k = 2$, heights experiment

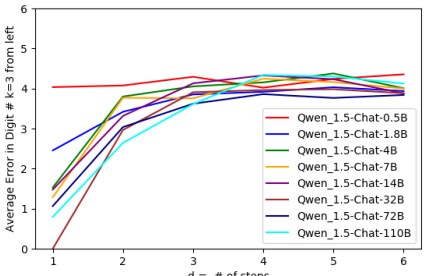

Figure 19: Average error vs # steps, digit # $k = 3$, heights experiment

