# OpenReview forum: "Regularity explains emergence"
_ICLR.cc/2025/Conference — ICLR 2025 Conference Withdrawn Submission_

### Official Review · Reviewer_hiKf · 2024-11-01

**Soundness:** 2
**Presentation:** 1
**Contribution:** 2
**Rating:** 3
**Confidence:** 3

**Summary:**

This paper studies emergence of capabilities as a function of model size. It tries to argue that "emergence" happens in cases where the
derivative of the ground truth is very large, and where larger models manage to approximate better. Experiments are provided 1) running a series of  ResNets on a small domain sin/cos function and 2) querying Qwen models on multiplication, addition, and language-formulated multi-step addition.

**Strengths:**

The idea to try to relate expressiveness of the model in terms of its ability to approximate steeper functions could be interesting,
if developed well.

**Weaknesses:**

To begin with, the paper is very badly written, to the point of being unreadable. I had to apply a lot of guessing goodwill to try to understand what the main claims are. To illustrate with one example: The statement of Theorem 2.5 says "Under assumptions 2.4" (this is an assumption on the boundedness of the difference between loss of the minimizer and loss of the parametric minimizer with N parameters, called (5) in the paper) "... Instead of the upper bound (5), which yields an infinite value...". How can you assume a finite bound and then say it's infinite?
Sadly, the paper is so full of defectuous English that even with the best of interpretations it is not possible to follow beyond the vague main ideas.

It is not really clear what the contributions are on top of the work cited: Siegel and Xu,20, E et al 22, Wu 2023. Beyond a combination of results, what is new?
Surely, it is known that large variability of the ground truth around a point gives more trouble to a model, and larger models interpolate better.

Another weakness pertains to the experiments. It is difficult to see how they illustrate the theoretical claim. First, a lot of assumptions are being made on the derivative, which is the key object of study. Like line 273: "we will assume that [the embedding] keeps the metric space structure of the set {0, 1, · · · , 9}" - without any justification I don't see why this is true. It is not clear to me how Figs. 1 and 2 demonstrate *any* emergence (and error bars are completely missing everywhere).
For the Qwen based LLM experiments, I am surprised how small the dataset is (128?).
There might be a potentially interesting observation in Lemma 3.2 saying that derivatives of middle digits are larger and thus harder to learn for small models, but the way this is written it is unclear whether this is true, and there might be confounding issues here (for instance, it's easy to guess whether the last digit is even or odd, given the two numbers to multiply; it could be that allowing 0 for the first digit increases the probability that guessing 0 there is correct, etc etc).
Figs. 3 and 4 are not very conclusive without error bars, especially for such small training sets.

This paper needs to be carefully rewritten (and it wouldn't hurt to use a language model for grammar control). Apart from grammatical errors, there is general sloppiness (for instance,  line 424 goes from "Grace" to "Tina", to name just one of many examples. Or the missing definition of "i" in the sum in line 219. What are d and k in line 219... Etc. )
The color scheme on all figures should be unified to go from lighter to darker (or something like that) for larger models - it doesn't help to have a color mix.

**Questions:**

1) I am unclear on what exactly your contribution is and what was already implicit in prior work - can you make that precise?
2) Why is the assumption that embeddings of digits preserve the metric space structure true?
3) Why are your datasets so small? What are the error bars? What am I supposed to see in the Figures ?
4) What are the bars above the variables starting line 282?
5) Sec 4.2. CoT: can you say anything more specific beyond speculation? Why are the derivatives multiplying and reducing?
6) Why do you need a 2-component (2dim) function in your first example line 219 (why is one component not enough here?)?

---

> ### Author Response · Authors · 2024-11-28
>
> - Response to Q1 : We prove that the optimal policy for the model is to give up difficult tasks. So it is "Prediction vs No prediction" instead of "Good prediction vs Bad prediction". Another way to put it is that the universal approximation works cited assume bounded Sobolev norms; the proposal of this paper is that the model cuts off unbounded Sobolev norms at a suitable threshold depending on model capacity to become bounded.
>
> - Response to Q2: We believe for optimization of mathematical tasks, it should be true for well-trained embeddings in a quasi-isometric sense that $1$ is closer to $2$ than to $9$. We will verify it in future versions by inspecting the embedding.
>
> - Response to Q3: We will pay attention to more detailed experiment results in future versions.
>
> - Response to Q4: The overline stands for decimal concatenation. By $\overline{456}$ we mean the decimal integer 456.
>
> - Response to Q5: Chain Rule
>
> - Response to Q6: 1D is enough. We used 2D to better demonstrate the phenomenon that the prediction is at the origin when the model doesn't have enough capacity to predict.

---

### Official Review · Reviewer_4byG · 2024-11-02

**Soundness:** 3
**Presentation:** 2
**Contribution:** 2
**Rating:** 5
**Confidence:** 4

**Summary:**

This paper proposes an explanation for the mechanism behind emergent capabilities in large language models through the regularity of the optimal response function. The authors claim that models do not model the optimal response in regions where derivatives are large, instead opting to predict a smoother function obtained through averaging values. They justify this theoretically and have accompanying experimental results on a synthetic function and certain arithmetic tasks (multiplication, sequence of single-digit addition, and addition word problems), where some intuitions from their theory are reflected in the accuracy trends of Qwen models as the number of parameters scale.

**Strengths:**

The theory is presented clearly, and the perspective of parameter size controlling the threshold on the extent to which the model predicts an irregular optimal response function is an interesting idea. The experimental setups are clear, and the synthetic setup is particularly compelling.

**Weaknesses:**

- While the theory seems sound and the synthetic experiment is compelling, I still reserve some skepticism for the connection to the LLM experiments on arithmetic tasks. Particularly I believe the title of this paper “Regularity Explains Emergence” is very strong and has a high burden of proof especially given the numerous natural language tasks where emergence has been observed [1] and the extensive discussions around the empirical validity of emergence in the existing literature (eg. [2]).
- To expand on this point, I currently can’t disentangle whether the theory provided by the authors truly gives an explanation for emergent capabilities in LLMs as they claim, or it provides one instance where emergence can occur and one can frame a theoretical narrative around. For the arithmetic tasks, while I can see that there can be conclusions drawn from the approximations of the gradient vector that are reflected in the accuracy trends across model scales and quantities like digit position and number of summands, I’m not convinced this is a result that we wouldn’t already expect intuitively and is necessarily explained from the theoretical results. The causal connection is not strong, likely due to the limitations of the theory and how it cannot explain more nuanced trends in eg. model scale (please see Questions below for expansion on this point).
- In conclusion, I believe that the authors need to be more clear about the scope of their theory and the tasks considered in this work, or provide stronger connections between the observed emergence and the regularity of the optimal function. Are there examples of natural language tasks where the theory may predict a regular optimal response function and we do see linear improvements in the task across scale?

- As a minor comment, there are areas in the paper where the writing has some typos and grammatical errors; I’ve listed several below but I’d like to ask the authors to go over their exposition and address some of the writing.

Line 19: improves -> improve

Line 44: \citet instead of \citep

Line 47: task -> tasks

Line 53: (Theorem 2.5 ->  (Theorem 2.5)

Line 56: avilable -> available

Line 58: LLM model -> LLMs

Line 61: method -> methods

Line 282: and R-value function -> an R-value function

Line 391: Figures 6-6 -> Figures 7-8

Line 391: despite of -> despite

Line 482-483: “On the other hand…” sentence needs rephrasing

[1] Srivastava, Aarohi, et al. "Beyond the imitation game: Quantifying and extrapolating the capabilities of language models." arXiv preprint arXiv:2206.04615 (2022).
[2] Schaeffer, Rylan, Brando Miranda, and Sanmi Koyejo. "Are emergent abilities of large language models a mirage?." Advances in Neural Information Processing Systems 36 (2024).

**Questions:**

1. Do you have any insights about how the relation between the number of parameters N and the optimal \epsilon(N) is reflected in the accuracy plots for the arithmetic tasks? For instance, it seems that the threshold ‘saturates’, in the sense that for 32B-110B the accuracy is similar even for the digits where accuracy is not at 100%. As a visualization, could you show what the accuracy looks like for a fixed digit position and x-axis being model scale (from Figures 3-4)?
2. You present average error results in Appendix C for the arithmetic tasks, and while general trends are the same as accuracy, it seems much noisier and the trend is not as consistent across model scale (eg. similar error between models with a difference of 2 orders of magnitude). Do you have any explanations for this, and could you also report the standard error across examples for the average error results?
3. What was the reasoning behind choosing summation of single digit integers as opposed to performing regular addition on d digits, analogous to the multiplication setting? How would the results change for potentially ‘harder’ or ‘simpler’ subsets of examples on these arithmetic tasks (for example, addition where there’s no carry for the first digit, or multiplication where there’s no carry across the digits?)
4. From the Big-Bench paper it was shown that the tasks exhibiting the most ‘linear’ trend in performance were perhaps more knowledge-based or required easier text manipulations, and the tasks with more ‘breakthrough’ performance trends had logical reasoning/sequential steps. How would this relate under your framework? I’m not sure these differences in the tasks are necessarily reflected in the regularity of the optimal function.

---

> ### Author Response · Authors · 2024-11-28
>
> Many thanks for your careful reading and detailed comments. Your opinions are greatly helpful for us to better approach the problem. Thank you for appreciating the point of using a ResNet setup to demonstrate the phenomenon before generalizing to the setting of an LLM. We agree with you that a general justification of the theory would be a high burden, and will work on it, especially in the natural language domain.
>
> We also thank you for finding typos and grammatical errors. We will correct them in future versions.
>
> Response to Q1 : Thank you for this question which hits right to the point. It is theoretically possible to make this estimate by going through the proof of Theorem 2.5 and the main ingredient of the resulting estimate will be the tail shape of the distribution of derivatives at all input-output pairs in the training dataset of the LLM. The fact that this distribution is not publicly available and expensive to compute, even if the training data is fully accessible, makes it hard to make a precise estimate. We believe that the fact that accuracy saturates is explained by the fact that when difficulty (regularity) of task increases, that is,
>  decreases, (1) the amount of available training data at that level of difficulty dramatically decreases, (2) the scaling law requires N to scale polynomially to adapt. Combining the two means a moderate drop in
>  will require a significant upscaling of
>  to compensate, i.e. the scaling law ends and one needs other strategies such as CoT as alternatives. We will spend time to work on a new version and are not uploading a new version this time, but can confirm that a plot as what you suggested does show such saturation.
>
> Response to Q2 : We hypothesize that beyond a regularity threshold, all models give up and provide random answers, resulting in similar but noisy error levels, which would be consistent with our interpretation.
> The standard error of error are roughly proportional to mean error and displays the same saturation pattern. For example, in experiment from 3.4, for the digit k=3, the standard error is almost constant, very close to 2 across all models and all number of steps >= 3, implying the the models are returning the randomly distributed responses beyond that regularity level.
>
> Response to Q3 : The reason of using summation of single digits is that the regular digit addition is easier from our regularity view point, in the sense that a digit in the answer likely only depends on adjacent digits from the inputs, while the summation of multiple single digit numbes has multiple dependencies. For the two examples you mentioned, the first one is a very restrictive subset of summation of multiple single digit numbers and below the bar to see interesting emergence. For the second one, the carry matters only slightly and regularity still grows fast with number of digits and we still see the U-shape.
>
> Response to Q4 : Thank you for suggesting the contrast, the "linear trend" type represents insufficiency of training data and the "breakthrough" type represents high regularity of optimal response and is closer to our framework. This aligns with our hypothesis in the following way: Problems that need logical reasoning/sequential steps can be viewed as a composition of problems. The optimal function is thus a composition of a sequence of optimal functions. By chain rule, the derivative of a composition of functions is equal to the product of individual functions' derivatives. Thus, the magnitude of its derivative is much bigger than that of a single function. Knowledge-based questions are either an individual function or a sum of these functions. Therefore, its derivative is of the same magnitude as that of an individual function.

---

> > ### Comment · Reviewer_4byG · 2024-12-03
> >
> > I thank the authors for their response. I have read the other reviews and responses and I will keep my score; as the authors have mentioned, the causal link between their proposed framework and experiments need to be strengthened if they want to support more general claim— otherwise, a reframing about the scope of the paper is needed.
> >
> > I appreciate the author’s response to my questions and understand that it would be difficult to derive precisely, but an estimate that can predict the model scale at which breakthroughness occurs, on any task, would be compelling. Right now, the theory gives some implications for some looser trends that should be observed with the relevant quantities (eg. k, d). However, I do think this is an interesting perspective where things like fine-tuning, chain of thought, and the difficulty of tasks exhibiting ‘linearity’ can have an intuitive explanation through looking at regularity. It might be interesting to include experiments even in simpler settings which do exhibit the contrapositive, eg. The authors claim that tasks which do not exhibit emergence must be highly regular.

---

### Official Review · Reviewer_wYtf · 2024-11-03

**Soundness:** 2
**Presentation:** 2
**Contribution:** 2
**Rating:** 3
**Confidence:** 3

**Summary:**

Paper introduces the idea that LLMs and other machine learning display a smoothing behavior in regions of input space where derivative of model output with respect to input is large. The behavior is said to emerge in specific regions of parameter space where training data has a “large derivative” in such regions of input space the result is that the network learns a “smoothed version” of the input output map rather than the map itself. The
claim is that the averaging behavior scales with parameters number and can yield to “emergence”-- where performance of model jumps on specific tasks as a function of parameter number. The authors introduce and prove a theorem which states that when a model, neural network map, cannot meet a performance standard within epsilon, then the model will learn an averaged version of the training data. The paper then provides numerical experiments with ResNet for fitting a trigonometric function and then uses the Qwen chat model for some analysis of algebraic operations.

**Strengths:**

I found the central claim interesting but preliminary for several reasons. Theoretical insight into how computations in language models can achieve zero shot task behavioral changes– for example– sorting a last in ascending vs descending order based on small changes in prompt are interesting. The idea that behavior on such tasks is influenced by the magnitude of local derivative of output on training data leading to learning of an averaged function are interesting - -although it isnt clear how the smoothed function can perform computations insight clear.

**Weaknesses:**

Technically, I find the notion of derivative in token space to be problematic. I have worked on similar problems in the case of CNNs where the notion of the derivative is well defined because inputs can be taken to be over the real numbers.

The problem with prompts is that tokenization causes the input domain for networks to be discrete valued (say integer valued), and the nature of the derivative on such input spaces is more more subtle. How is the derivative to be defined on such spaces? The problem is that the local behavior of a derivative taken on Z embedded into R is not representative of the notion that the authors seek– which is a function that measured changes on input instances.

Therefore, I would like to see a much more rigorous development of the main theorem with specific definition and analysis of the derivative for token valued functions which are the main object of study for LLMs.


Second, the numerical experiments in the paper are very limited– the title of the paper is about language models, but the first experiment is on ResNet.

The language model experiment is limited and I do not see a global investigation of this notion of the network derivative in different regions of parameter space and the input-output function f or the “smoothed version S*f.

Can the authors systematically evaluate the derivative and inferred the smoothed input-output function on a more general class of language models?

To solidify their central claim, can the authors analyze models of increasing size showing convergence to their central claim with model size?

**Questions:**

How do the authors define the derivative over token valued neural networks?

Can the authors systematically evaluate the derivative and inferred the smoothed input-output function on a more general class of language models?

To solidify their central claim, can the authors analyze models of increasing size showing convergence to their central claim with model size?

---

> ### Author Response · Authors · 2024-11-28
>
> Thank you very much for the insightful comments.
>
> -Response to "I would like to see a much more rigorous development of the main theorem with specific definition and analysis of the derivative for token valued functions which are the main object of study for LLMs."
>
> We appreciate this question and wish to hear more of your insight. Our assumption is that an appropriate tokenization is an optimally smoothified embedding of a discrete natural dictionary, in the sense that words with similar meaning are closer to each other. And there are many dimensions to represent similarity in different aspects. In fact, the subsequent layers of an LLM are smooth neural networks that treat the tokenized inputs continuously. The success of these models itself implies that the smoothness of the tokenization. Please share your opinion on this view.
>
> -Response to "the numerical experiments in the paper are very limited"
>
> The phenomenon of the numerical experiments in the Resnet sheds light on the fundamental philosophy of the issue that we are discussing. While we agree it is a toy experiment, we would like to include it to provide the first evidences. We plan to design more experiments in an LLM setting.
>
> -Response to "The language model experiment is limited"
>
> We agree that the experiments only covers a very special type of tasks and are short of showing the full landscape of the function $f$. The main obstruction is, indeed as you ask in the next question, the difficulty in systematically evaluate the derivative of $f$.
>
> -Response to "How do the authors define the derivative over token valued neural networks?"
> As mentioned above, we think token valued inputs should be viewed as continuously valued once they are embedded as vectors in the sense that nearby vectors have similar meanings. But we are open to hear different opinions on this issue.
>
> -Response to "Can the authors systematically evaluate the derivative and inferred the smoothed input-output function on a more general class of language models?"
>
> We acknowledge that this is hard and in fact it is what limited us to arithmetic tasks whose derivatives are more explicit. We can think of evaluation approaches on some other task classes but they are still partial and not systematic enough. Any advice or references will be appreciated.
>
> -Response to "can the authors analyze models of increasing size showing convergence to their central claim with model size?"
>
> We are not sure if we understand this question but assume that you are asking about making the convergence more effective, or the relations between $\epsilon(N)$ and $N$. The short answer is yes on the theoretical level. But the longer answer is that the quantitative estimates requires knowledge about the tail distribution of derivative norms of training data used by the LLM, which is not accessible.

---

### Official Review · Reviewer_tDVV · 2024-11-12

**Soundness:** 3
**Presentation:** 2
**Contribution:** 2
**Rating:** 5
**Confidence:** 4

**Summary:**

This paper investigates the concept of "emergent abilities" in large language models (LLMs) by developing a theoretical framework based on the regularity (or smoothness) of the optimal response function. The authors suggest that LLMs approximate this response function by smoothing out regions with high derivative values, leading to approximation errors that gradually decrease as the model size, N, grows. The theory proposes that as N increases, the model can capture more complex aspects of the response function without the need for smoothing, which results in sudden improvements or "emergence" of new abilities. The authors present a key theorem that quantifies the relationship between model size and approximation quality. They also provide experimental evidence to support the theory, including function approximation with ResNets and arithmetic tasks to demonstrate the model’s behavior in regions with high derivatives.

**Strengths:**

- I appreciated the theoretical framework based on Siegel & Xu (2020), which links the regularity of optimal response functions with the concept of emergence. This framework offers a fresh perspective on the phenomenon of "emergent abilities" in large language models (LLMs).

- The main theorem effectively illustrates how model size relates to approximation quality, especially in regions where the optimal response function shows complex behavior. Although primarily qualitative, this theoretical foundation provides valuable insights into why larger models may perform better with irregular functions.

- I found some of the empirical results intriguing, particularly the scaling experiments with Qwen models that revealed various trends in arithmetic calculation outcomes.

**Weaknesses:**

- Although the paper provides a unique and intuitive perspective on the mechanisms underlying emergence, it doesn’t specify a precise threshold or clear scaling rule to predict when this emergence occurs. I would appreciate it if the authors could better highlight exactly what constitutes the list of "quantitative/concrete" predictions proposed by the theory.

- The toy model experiments using ResNet don’t closely match the large language models (LLMs) setup. This setup is qualitatively different from the autoregressive transformers typically used in LLMs. While the authors argue that the theory applies to any model type, this actually highlights a limitation of the theory rather than supporting the use of ResNets to examine phenomena observed in LLMs.

- The choice of arithmetic tasks doesn’t clearly connect to the theory’s focus on changes in derivatives, as the observed U-shaped trend can be explained solely by the task structure. Choosing tasks more closely aligned with the theory would make the paper’s ideas clearer and more applicable.

- Overall, while I find the results in some parts of the paper interesting, they often appear disconnected, lacking a clear and logical progression.

- Presentation should be improved. In particular, it would greatly help if captions contained the necessary information to understand the content beyond what is provided in the existing title headers.

**Questions:**

- Can the theory predict specific model sizes or conditions where emergent behavior happens? Is there a certain size, N, where the model shifts from smoothing f∗ to accurately capturing it in areas with sharp changes?

- Could you design an experiment with an autoregressive transformer model that would produce results more relevant to the theory?

- Can the theory’s error bounds predict error rates for specific tasks at different model sizes?

---

> ### Author Response · Authors · 2024-11-28
>
> Thank you very much for the insightful comments.
>
> -Response to "..  highlight exactly what constitutes the list of ``quantitative/concrete" predictions proposed by the theory."
>
> The way to impose threshold of emergence is when the slope of accuracy curve is changing. There are significant slope change in the plot. In a future version, we will add a description to address this point.
>
> -Response to ".. The toy model experiments using ResNet don’t closely match the large language models (LLMs) setup. "
>
> We are optimistic that our interpretation is universal but open to other possibilities, and we accept the fact that ResNet is very different from LLMs. We would prefer to think this is rather a limitation in our experiments rather than of the theory, but would love to learn about why a priori LLMs and ResNet would require intrinsically different mechanisms for emergence.
>
> -Response to "Choosing tasks more closely aligned with the theory would make the paper’s ideas clearer and more applicable."
>
> We chose arithmetic tasks as it is mathematically possible to compute derivative of the target function for them. For a real-world question, for instance,
> "What is Shakespeare's most famous masterpiece?" it would be very difficult to analyze or even estimate the size of derivative of the target function. With tokenization and embedding, the function may be distorted in a very complicated way, and the solution's implicit relationship to the variation of the input is hard to measure. We would appreciate insights on this obstruction to the implementation of more profound experiments.
>
> -Response to "Can the theory predict specific model sizes or conditions where emergent behavior happens?"
>
> This is a very relevant question and we appreciate it. We believe this question is closely related to the next question you ask, as well as the relation between $N$ and $\epsilon(N)$ that reviewer 4byG asks about. It is theoretically possible to make this estimate by going through the proof of Theorem 2.5 and the main ingredient of the resulting estimate will be the tail shape of the distribution of derivatives at all input-output pairs in the training dataset of the LLM. The fact that this distribution is not publicly available and expensive to compute even if the training data is fully accessible makes it hard to make a precise estimate.
>
> -Response to "Could you design an experiment with an autoregressive transformer model that would produce results more relevant to the theory?"
>
> As we mentioned above, the main difficulty lies in determining the derivative norms of the ground truth target function, which made us stick to ResNet and arithmetic tasks in current experiments. We would appreciate any advice on the design of that part of the experiment.
>
> -Response to "Can the theory’s error bounds predict error rates for specific tasks at different model sizes?"
>
> In a perfect world where the tail distribution of ground truth derivatives is known, the proof of the main theorem would allow to predict the relation between $N$ and $\epsilon(N)$, namely at what difficulty level the model starts to give up tasks, which would cover part of the error rate. The remaining error rate, i.e. the chance that the model does make an effort but fails, is separate and can be estimated by the Siegel-Xu bound more directly.}

---

### Author Response · Authors · 2024-11-28

We thank the reviewers for their careful reading and valuable opinions. We accept the main criticism that more experiments beyond arithmetic tasks are needed to support general causality, and explain some obstructions to that goal in our responses below. We will spend more time on designing experiments to support our interpretative frame work and write a new version, and are not uploading a revision for now. We post our responses below and will be happy to receive more advice from the referees in the next few days.

---

### Note · Authors · 2024-12-11

**Comment:**

We sincerely thank all reviewers for their careful reading and valuable opinions. We will continue the next stage of our research with these suggestions in mind.

**Withdrawal Confirmation:**

I have read and agree with the venue's withdrawal policy on behalf of myself and my co-authors.